# Screening of Neutralizing Antibodies against FaeG Protein of Enterotoxigenic *Escherichia coli*

**DOI:** 10.3390/vetsci11090419

**Published:** 2024-09-09

**Authors:** Yang Tian, Sijia Lu, Saisai Zhou, Zhen Li, Shuaiyin Guan, Huanchun Chen, Yunfeng Song

**Affiliations:** 1National Key Laboratory of Agricultural Microbiology, Huazhong Agricultural University, Wuhan 430070, China; tian_1995@webmail.hzau.edu.cn (Y.T.); lsj20010712@163.com (S.L.); zss2021302010119@webmail.hzau.edu.cn (S.Z.); swanlizen@webmail.hzau.edu.cn (Z.L.); guanshuaiyin@webmail.hzau.edu.cn (S.G.); 2College of Veterinary Medicine, Huazhong Agricultural University, Wuhan 430070, China; chenhch@mail.hzau.edu.cn

**Keywords:** enterotoxigenic *Escherichia coli*, soluble rFaeG protein, fluorescence-activated cell sorting, IPEC-J2, neutralizing antibody

## Abstract

**Simple Summary:**

This study focused on developing a monoclonal antibody against the F4ac subtype of the FaeG protein of enterotoxigenic *Escherichia coli* by screening an antibody library in immunized mice, providing a potential drug option for antibiotic replacement therapy in veterinary medicine. The researchers immunized mice with formaldehyde-inactivated ETEC and a soluble recombinant FaeG protein to generate an antibody library. Through fluorescence-activated cell sorting and a eukaryotic expression system, we screened and obtained anti-rFaeG IgG monoclonal antibodies that effectively inhibited ETEC’s adhesion to IPEC-J2 cells. In vivo experiments confirmed the protective effect of these antibodies in mice challenged with ETEC, demonstrating their potential as a safe and effective therapeutic option.

**Abstract:**

The misuse of antibiotics in veterinary medicine presents significant challenges, highlighting the need for alternative therapeutic approaches such as antibody drugs. Therefore, it is necessary to explore the application of antibody drugs in veterinary settings to reduce economic losses and health risks. This study focused on targeting the F4ac subtype of the FaeG protein, a key adhesion factor in enterotoxigenic *Escherichia coli* (ETEC) infections in piglets. By utilizing formaldehyde-inactivated ETEC and a soluble recombinant FaeG (rFaeG) protein, an antibody library against the FaeG protein was established. The integration of fluorescence-activated cell sorting (FACS) and a eukaryotic expression vector containing murine IgG Fc fragments facilitated the screening of anti-rFaeG IgG monoclonal antibodies (mAbs). The results demonstrate that the variable regions of the screened antibodies could inhibit K88-type ETEC adhesion to IPEC-J2 cells. Furthermore, in vivo neutralization assays in mice showed a significant increase in survival rates and a reduction in intestinal inflammation. This research underscores the potential of antibody-based interventions in veterinary medicine, emphasizing the importance of further exploration in this field to address antibiotic resistance and improve animal health outcomes.

## 1. Introduction

Enterotoxigenic *Escherichia coli* (ETEC) is the primary pathogen responsible for *E. coli*-associated diarrhea in piggery [1]. ETEC gains entry into and colonizes the small intestine via fimbriae, targeting specific receptors on the intestinal epithelium [2,3]. Following colonization, ETEC firmly adheres to the intestinal epithelium, facilitating the synthesis and secretion of multiple enterotoxins [3,4]. Similar to its transmission pathways in humans, ETEC can survive for extended periods within fecal matter and spread via the fecal–oral route within piggeries [5,6]. This transmission leads to the onset of yellow and white scour in newborn piglets, resulting in increased morbidity and mortality rates, reduced piglet viability, and significant economic losses in the swine industry.

The F4 fimbriae of ETEC, also known as K88, are among the most critical adhesion factors of ETEC [7]. The principal subunit of these fimbriae, the FaeG protein, is externally exposed within the F4 fimbriae structure and plays a central role in supporting and stabilizing the entire fimbrial architecture [8,9,10]. Additionally, the FaeG protein is readily recognized by the host immune system, eliciting a protective immune response [11,12]. Although the FaeG protein exists in three antigenic variants—F4ab, F4ac, and F4ad—the F4ac variant is the most prevalent in piggery environments [13,14]. Therefore, the FaeG protein of F4ac–ETEC holds promise as an immunogen for developing prophylactic and therapeutic biological products targeting ETEC.

Moreover, the misuse of antibiotics in agricultural settings has become a pressing public concern. Excessive antibiotic usage can result in residues persisting in pork, posing potential risks to human health [15,16,17,18]. Prolonged and heavy antibiotic administration fosters bacterial resistance, thereby compromising treatment efficacy and exacerbating challenges in disease control and prevention. From an environmental standpoint, antibiotics may leach into the environment through pig feces, leading to soil and water pollution and disrupting the ecological equilibrium [19]. Consequently, numerous countries have initiated measures to promote the adoption of antibiotic alternatives for animal disease prevention and treatment. Among these alternatives, antibody drugs represent a promising avenue of research, utilizing monoclonal antibodies (mAbs) or their derivatives as therapeutics. Each immunoglobulin monomer is a symmetric heterotetramer, consisting of two heavy chains and two light chains (H2L2). The light chain (LC) is covalently bonded to a heavy chain (HC) via disulfide bonds. Furthermore, the antigenic complementary determining regions that they form are the most crucial sites of the antibody and are frequently targeted for research. Currently, they find widespread application in treating human cancers, infectious diseases, and autoimmune disorders [20]. Concurrently, advancements in molecular cloning technology and flow cytometry have streamlined the screening and production of antibody drugs, rendering the process highly targeted and time-efficient. Furthermore, the evolution of bioinformatics and genomics has ushered in a plethora of personalized clinical treatment options [21].

However, there are currently limited studies on antibody drugs for animals. Therefore, exploring the application of antibody drugs in veterinary medicine is imperative to mitigate economic losses and the adverse effects of antibiotic misuse. Currently, although the cost of antibody-based drugs is significantly higher than that of antibiotics, research into antibody derivatives is emerging [22,23]. This includes both eukaryotic and prokaryotic expression systems. Optimizing these expression conditions could gradually reduce the production costs of antibody therapeutics. In this study, we targeted the F4ac subtype of the FaeG protein. Formaldehyde-inactivated ETEC and a soluble recombinant FaeG (rFaeG) protein were utilized to establish an antibody library against the FaeG protein. This was combined with fluorescence-activated cell sorting (FACS) and a eukaryotic expression vector containing murine IgG Fc fragments to screen for anti-rFaeG IgG mAbs. In terms of experimental models, compared with mouse models, piglets are more expensive and have complex husbandry requirements, making them less suitable for preliminary studies. Additionally, there are established reports on ETEC-induced enteritis models in mice [24]. An intraperitoneal injection of ETEC can cause symptoms such as rough fur, body hunching, diarrhea, and even death in mice. Therefore, we utilized a mouse model, which is easier to handle and has more controllable environmental conditions, for preliminary research. The results demonstrate that the screened variable region of the antibody could effectively inhibit the adhesion of ETEC to IPEC-J2 cells, thus providing a safe and efficient antibody drug for the treatment of neonatal piglet diarrhea.

## 2. Materials and Methods

### 2.1. Median Lethal Dose and Minimum Lethal Dose Test of Isolated ETEC Strains

In order to determine the dose required to induce immunity in mice to serum antibodies and verify the neutralization capacity of the antibodies for subsequent experiments in mice, it was necessary to determine the median lethal dose (LD_50_) and the minimum lethal dose (MLD) of the isolated ETEC strain (a strain of the K88 type was isolated from pigs in Guangxi, China). The LD_50_ was used for neutralization assays in mice, and the MLD was used for the immunization of mice. We designed a density gradient of ETEC in sterile PBS, including 0.5 × 10^7^ CFU/mL, 1 × 10^8^ CFU/mL, 2 × 10^8^ CFU/mL, 4 × 10^8^ CFU/mL, 8 × 10^8^ CFU/mL, and 1 × 10^9^ CFU/mL [25]. A total of 28 mice were randomly divided into seven groups and injected intraperitoneally with 200 μL of a sterile PBS suspension of ETEC after 24 h of fasting. After 72 h of challenge, the death of the mice was observed to determine the LD_50_ and the MLD [25,26].

### 2.2. Immunization of Mice

The mouse experiment was conducted in accordance with the guidelines for the care and use of laboratory animals at Huazhong Agricultural University. The procedures for the animal experiments were approved by the Science Ethics Committee of Huazhong Agricultural University (Approval Number: HZAUMO-2024-0171). Female Balb/c mice were purchased from the Laboratory Animals Centre of Huazhong Agricultural University and housed under specific pathogen-free conditions. Five mice were randomly selected for immunization, numbered 0–4#, and they were immunized with a mixture of 0.3% formaldehyde-inactivated ETEC (MLD) and alum adjuvant (Thermo Fisher Scientific, Waltham, MA, USA), except for mouse 0#, which remained seronegative without immunization. Blood samples were collected from the orbital venous plexus the day before each immunization. Each mouse received a multipoint subcutaneous injection in the neck and back (150 μL per mouse), with immunizations administered every other week. After three immunizations, 200 μL of inactivated ETEC solution (MLD) without adjuvant was injected intraperitoneally to enhance immunity. Three days later, the mice were anesthetized with isoflurane, and blood samples were collected via the orbital venous plexus before each treatment. All mice were ultimately euthanized via an intraperitoneal injection of an overdose of pentobarbital sodium (100 mg/kg).

### 2.3. Expression and Fluorescent Labeling of F4ac FaeG Protein in ETEC

To amplify the *faeG* gene without a signal peptide, according to its Genebank ID CP148324.1, we used the whole DNA genome of ETEC as a template. The amplification was carried out using 2 × Fastpfu Fly PCR SuperMix (TransGen Biotech, Beijing, China) with the following primers: *faeG*-F-*Nde*I (GGGAATTCCATATGACTGGTGATTTCAATGG) and *faeG*-R-*Xho*I (CGGCTCGAGTTAGTAATAAGTAATTGCTACGTT). The PCR protocol consisted of an initial denaturation at 95 °C for 1 min, followed by 35 cycles of denaturation at 95 °C for 10 s, annealing at 51 °C for 15 s, extension at 72 °C for 10 s, and a final extension at 72 °C for 1 min, with storage at 4 °C. Subsequently, the pET-28a-FaeG plasmid was constructed using the expression vector pET-28a (Merck, Darmstadt, Germany), and it was co-transfected into *E. coli* BL21(DE3), along with the chaperone expression vector pGro7 (Takara, Tokyo, Japan). The *E. coli* wascultured in Luria–Bertani (LB) medium supplemented with chloramphenicol (20 µg/mL), kanamycin (50 µg/mL), and L-arabinose (2 mg/mL) at 37 °C and 220 rpm until the bacterial suspension reached an optical density at 600 nm (OD_600_) of approximately 0.6. Protein expression was then induced by adding 1 mm isopropyl β-D-thiogalactoside (IPTG) and incubating the culture at 16 °C with agitation at 160 rpm for 16 h. Finally, the bacteria were disrupted with high pressure and centrifuged to collect the supernatant at 134,000× *g* in 4 °C.

Post-expression, the protein was purified using Ni Sepharose 6 Fast Flow (GE Healthcare Biosciences, Piscataway, NJ, USA), achieving a purity of over 90%. The buffer was exchanged for PBS using a 10 kDa ultrafiltration concentration tube (Merck, Darmstadt, Germany). The concentration of the rFaeG protein was measured using a BCA Protein Assay Kit (Beyotime Biotechnology, Shanghai, China). Finally, the purified protein was labeled using an APC Conjugation Kit and a PE/R-Phycoerythrin Conjugation Kit (Abcam, Cambridge, UK).

### 2.4. Indirect Enzyme-Linked Immunosorbent Assay (iELISA)

The FaeG protein was diluted to a concentration of 2 μg/mL in a carbonate buffer (pH 9.6) [Na_2_CO_3_ 15 mm, NaHCO_3_ 35 mm] or coated with 4 × 10^9^ CFU/mL of living ETEC at 4 °C 100 μL/well overnight. Additionally, the primary antibody was added at 1:100. Then, HRP-labeled goat anti-mouse IgG (Jackson Immuno Research Laboratories, Philadelphia, PA, USA) was added at 1:5000. After washing, 3,3′,5,5′-tetramethylbenzidine (TMB) was added to each well, oxidized by HRP, and incubated at room temperature for 15 min away from light. The reaction was terminated by using 2 M sulfuric acid, and the OD_450_ value was measured using an ELISA reader (Tecan, Männedorf, Switzerland).

### 2.5. Fluorescence-Activated Cell Sorting

A splenocyte suspension was obtained from two immunized mice with high serum titers after grinding and filtering with a 70 μm cell strainer, washed with PBS buffer containing 5% BSA and 0.5 M ethylenediaminetetraacetic acid (EDTA), and then blocked with FcR-blocking reagent (Miltenyi Biotech, Bergisch Gladbach, Germany). B cells were then enriched using CD45R (B220) MicroBeads and LS MACS columns (Miltenyi Biotech, Bergisch Gladbach, Germany). After that, FITC anti-mouse IgG antibody (1 μg/10^8^ cells), AF700 anti-mouse CD19 antibody (2 μg/10^8^ cells), APC-FaeG, and PE-FaeG fluorescent markers were thoroughly mixed with cells away from light using a rotating device at 10 rpm and 4 °C for 30 min. Finally, the single-cell suspension was centrifuged at 4 °C 300× *g* for 10 min and then washed with buffer twice. The antigen-specific mouse memory B cells were sorted using a BD FACSAria™ III Cell Sorter (BD Biosciences, Franklin Lakes, NJ, USA) into 96-well plates containing 10 μL/well radio immunoprecipitation assay lysis buffer (RIPA, BeyotimeBiotechnology, Shanghai, China) with an RNase inhibitor, at one cell per well, and stored at −80 °C.

### 2.6. Amplification of Variable Region and Construction of Expression Plasmids

In this study, multiplex polymerase chain reaction (PCR) and nested PCR were used to amplify the antibody variable region of single B cells. The eukaryotic expression vector pcDNA3.4-mCγ/Cκ containing either the mouse Cγ or Cκ fragment was prepared and saved in the early stages in our laboratory. Single B cells were thawed on ice for 10 min, and then the first-strand cDNA was synthesized using a HiScript II 1st Strand cDNA Synthesis Kit (Vazyme, Ningbo, China) for the reverse transcription of each hole. The variable region of the light and heavy chains was amplified for the first time via a MyCycler™ Thermal Cycler (Bio-Rad, Hercules, CA, USA) using 25 μL 2 × Fastpfu fly PCR supermix (TransGen Biotech, Beijing, China). All primer sequences are shown in Table 1 [27]. The following PCR products were recovered using an E.Z.N.A.^®^ Gel Extraction Kit (Omega, Geneva, Switzerland). In the first PCR step, the forward primers (10 μM) containing the upstream homology arms related to pcDNA3.4-mCγ/Cκ were mixed in an equal amount. Then, the PCR products were taken for a 1% agarose gel electrophoresis analysis; according to this, the well with light- and heavy-chain target products was selected for a second amplification. The total PCR reaction system was 50 μL, and the annealing temperature was 62 °C. pcDNA3.4-mCγ/Cκ was linearized with *Xba*I restriction endonuclease. Next, 2 μL of the recovered products, 3 μL linearized vectors, and 5 μL of 2 × Uniclone Seamless Cloning Mix (Genesand Biotech., Shenzhen, China) were mixed and seamlessly cloned at 50 °C for 30 min in 96-well plates. Then, 25 μL *E. coli* DH5α and 150 μL LB-containing ampicillin (100 mg/mL) were added to each well, and the plates were then put on ice for 30 min and heat-shocked at 42 °C for 90 s. After that, the transformed DH5α was multiplied at 37 °C and 220 rpm until the liquid became turbid. Finally, the transformed DH5α was expanded, and the recombinant light- and heavy-chain plasmids were extracted with an Endo-Free Plasmid Mini Kit I D6948 (Omega, Geneva, Switzerland) and stored at −80 °C for later use.

### 2.7. Expression and Analyses of Anti-rFaeG IgG mAbs

It was ensured that the HEK293-6E cells grew to 3.0–5.0 × 10^6^ cells/mL before transformation, and recombinant light- and heavy-chain plasmids were simultaneously transfected 1:1 into HEK293-6E cells using Hieff Trans^®^ Liposomal Transfection Reagent (Yeasen Biotechnology, Shanghai, China). After five days of expression, the supernatant was collected to detect the expression of antibodies using the iELISA method. A pair of plasmids with the highest OD_450_ value was selected and transfected into 6-well plate cells again. After six days of culture, the supernatant was collected and placed on ice, and then it was slowly bound to the protein A column (Beyotime Biotechnology, Shanghai, China). The anti-rFaeG IgG mAbs were eluted with twice the volume of glycine-HCl buffer solution (pH 3.0). Then, the eluted antibody was concentrated and replaced by PBS with a 50 kDa Ultra Centrifugal Filter (Merck, Darmstadt, Germany), and the protein concentration was determined via the BCA method.

The purified mAbs were detected correctly via SDS-PAGE protein gel electrophoresis, and then the titer of the mAbs was detected via double dilution from 1:100, according to the iELISA method. In the specificity detection of anti-rFaeG IgG mAbs using Western blotting, DH5α, BL21, and non-F4-type *E. coli* were used for negative analysis, while the bacterial cells of isolated ETEC and rFaeG were used for positive analysis.

### 2.8. Adhesion Assay of ETEC

The functionality of the mAbs was verified via indirect immunofluorescence (IFA). ETEC was re-suspended and diluted to 3 × 10^8^ CFU/mL with PBS [28]. Then, we used the mAbs (10 μg/mL), positive serum, and negative serum (diluted 1:4 with PBS) to resuspend ETEC as three test groups; meanwhile, we set up an untreated ETEC control group. Then, all mixtures were incubated with a rotary mixer (Thermo Fisher Scientific, Waltham, MA, USA) at 37 °C for 1 h, except for the control group. When the IPEC-J2 cells were grown to 90% confluence with Dulbecco’s Modified Eagle Medium/Nutrient Mixture F-12 (DMEM/F-12, Servicebio, Wuhan, China) with 10% FBS in 96-well cell plates, 100 μL of each sample was added to each well, with three repetitions per group, and the cells were adhered at 37 °C for 30 min. After washing, precooled 100% methyl alcohol was added for fixation at room temperature for 30 min, and then 1% BSA blocking buffer was added at 37 °C for 1 h. The mice-immunized positive serum was diluted with a blocking buffer at a 1:400 dilution ratio as the primary antibody, and DyLight 594 Goat Anti-Mouse IgG H&L (Abcam, Cambridge, UK) was diluted with a blocking buffer at a 1:800 dilution ratio as the secondary antibody and incubated at 37 °C for 1 h away from light. Finally, 100 μL 300 nM 4′,6-diamidino-2-phenylindole (DAPI, Thermo Fisher Scientific, Waltham, MA, USA) was added to each well, and incubation was carried out at room temperature for 10 min away from light. An ECLIPSE Ti2-U inverted fluorescence microscope (Nikon, Tokyo, Japan) was used to observe and photograph.

### 2.9. Anti-rFaeG IgG mAb Neutralization Assay in Mice

A total of 15 female mice were randomly selected and divided into three groups, with each group comprising 5 mice. Group 1 received an intraperitoneal injection of LD_50_ of ETEC. In group 2, the same dose of ETEC was intraperitoneally injected after being incubated at 37 °C in sterile PBS with 10 μg/mL of mAbs for 1 h. Group 3 received an injection of the same volume of sterile PBS. Following injection, the survival of the mice in each group was monitored every 12 h, and the ileum tissues of the mice that succumbed to the infection were collected and fixed with 4% paraformaldehyde. The mice in groups 2 and 3 were euthanized with pentobarbital 72 h post-injection, and their ileum tissues were collected and fixed. Subsequently, the ileum tissues were embedded in paraffin blocks, sectioned, and stained with hematoxylin and eosin. Immunohistochemical (IHC) sections were prepared using the expressed mAbs as the primary antibody and HRP-goat anti-mouse IgG (Servicebio, Wuhan, China) as the secondary antibody for observation under a Motorized Fluorescence Microscope BX63 (Olympus, Tokyo, Japan).

## 3. Results

### 3.1. Soluble Expression and Immune Response of rFaeG Protein in Mice

In this study, the *faeG* non-signal peptide gene (786 bp) of the F4ac subtype was amplified and cloned into the recombinant plasmid pET-28a-FaeG. This plasmid was then co-transfected into BL21 (DE3) with the pGro7 plasmid. Finally, the recombinant FaeG was successfully expressed and purified in the supernatant, and its purity reached over 90% (Figure 1A). The protein concentration was determined to be 1.8 mg/mL using the BCA method.

The MLD determined for the mice was 4 × 10^8^ CFU/mL. After the immunization of these mice, ETEC and rFaeG were coated on ELISA plates to evaluate the immunogenicity of rFaeG using iELISA. The OD_450_ value of the serum samples increased with the immunization times. The results showed that the growth of positive antibodies in the serum detected by ETEC and rFaeG antigens was consistent (Figure 1B). Additionally, when the positive serum was double-diluted to 1:24,800 from 1:100, the OD_450_ value was still higher than that of the negative serum (Figure 1C), indicating that the FaeG protein, as the main subunit of the F4 fimbriae on the surface of ETEC, could effectively induce an increase in the immune serum titer in mice, and its conformation was more similar to that of natural fimbriae.

### 3.2. Sorting of rFaeG-Specific Mouse Memory B Cells

We selected two mice (1# and 2#) with the best immune effect to enrich B cells in the spleen using anti-CD45R (B220) MACS. The enriched B cells were incubated with FITC anti-mouse IgG antibody, AF700 anti-mouse CD19 antibody, and PE-/APC-labeled rFaeG. Then, the rFaeG-specific mouse memory B cells were analyzed using an FACS instrument (Figure 2). Finally, 58 rFaeG-specific memory B cells were sorted into a 96-well plate for single-cell culture.

### 3.3. Preparation of Anti-rFaeG IgG mAbs

Of the 58 single B cells selected, only 28 were successfully amplified in the IgG variable region of the heavy and light chains. These PCR products contained homology arms. Then, eukaryotic expression vectors for the light- and heavy-chain antibody variable regions were obtained through nested PCR and seamless cloning. Further, the 28 recombinant antibodies were expressed in HEK293-6E, and 21 wells were found to be positive after the detection of the cell supernatant using iELISA. We selected the pair of plasmids expressed in the B3 well with the highest OD_450_ value (Figure 3A) to prepare anti-rFaeG IgG mAbs (1.3 mg/mL). The SDS-PAGE results showed that the full lengths of the light and heavy chains were correctly expressed and combined into H2L2 (Figure 3B). Subsequently, the mAbs were further evaluated using iELISA. According to the results, the titer of the mAbs was significantly higher than 1:100,000 (Figure 3C). Additionally, it was found that only the FaeG protein had an antigen–antibody reaction with the mAbs, indicating the good specificity of the mAbs (Figure 3D). Furthermore, the eukaryotic expression vector of the mAbs was sent to Tsingke Biotechnology Co., Ltd. in Wuhan for sequencing. Then, we used a website “http://www.vbase2.org/ (accessed on 14 November 2023)” to blast the sequencing results and submitted them to NCBI. The GeneBank IDs of the heavy and light chains are PP824722 and PP824723, respectively.

### 3.4. Neutralizing Activity of Anti-rFaeG IgG mAbs In Vitro

To evaluate the suppression effect of the recombinant IgG antibody, we used ETEC and IPEC-J2 as research objects via the IFA method. ETEC was incubated with negative serum, positive serum, recombinant antibodies, and PBS. Then, the treated ETEC was infected with IPEC-J2 cells. From the results, in the negative serum and PBS groups (Figure 4A,D), a red fluorescence signal could be clearly observed, indicating that ETEC successfully adhered to the surface of the IPEC-J2 cells. However, when treated with the anti-rFaeG IgG mAbs (Figure 4C), the fluorescence signal significantly weakened, which was similar to the positive serum group (Figure 4B), indicating that the recombinant antibodies had a strong suppression effect on the adhesion of ETEC.

### 3.5. Neutralizing Activity of Anti-rFaeG IgG mAbs In Vivo

In the test of LD_50_, within 72 h of challenge, mice with different gradients showed symptoms such as coarse fur, curling up, and motility and fecal issues. After 72 h, the LD_50_ was determined to be 8 × 10^8^ CFU/mL through the statistics of dead mice. In the in vivo neutralization experiments, obvious differences could be seen by observing the three groups of mice after the intraperitoneal injection of ETEC only, after the injection of ETEC incubated with mAbs, and after the injection of sterile PBS. As shown by the survival curve (Figure 5A), it was evident that, after 72 h, only one mouse remained alive in group 1, which was injected with ETEC only, with the others dying sequentially. Conversely, in group 2, the mice injected with ETEC incubated with mAbs exhibited only one mortality event, with the remaining surviving. In group 3, the injection with sterile PBS did not result in any fatalities. Thus, after binding with ETEC, the antibodies significantly reduced mouse mortality in vivo. Moreover, the neutralizing effect of anti-rFaeG mAbs was further demonstrated through IHC and HE-stained tissue sections. As shown in Figure 5B, significant atrophy and damage to the ileal villi were observed in group 1 following the intraperitoneal injection of ETEC, leading to a loss of normal tissue architecture, with substantial inflammatory exudate present at the base and within the intestinal lumen. The IHC sections also revealed that the ETEC-positive brown–yellow staining in group 1 was markedly higher than in groups 2 and 3. In contrast, group 2, which received ETEC pre-incubated with mAbs, showed intact intestinal villi structures and no significant inflammatory exudate, with the ileal architecture under the microscope closely resembling that of group 3, which received sterile PBS as a control.

## 4. Discussion

Currently, research on recombinant antibodies is burgeoning, but it is primarily focused on human diseases, and the preparation and application of animal recombinant antibodies are scarce [29,30]. Moreover, ETEC is a common pathogen in piggery environments, posing significant threats to piglets. Typical treatment involves the use of antibiotics, which is not a sustainable solution. Although vaccination is effective, the preventive efficacy of vaccines is not yet satisfactory, leading to the continuous development of new vaccines [26]. However, the development of recombinant antibodies represents a novel approach to address the issue of antibiotic overuse in the pig industry. Virdi et al. successfully prepared anti-ETEC antibodies by fusing the variable structural domains of the camel heavy-chain antibody (VHH) with the Fc portion of porcine immunoglobulin (IgG or IgA) and expressing them in Arabidopsis thaliana seeds [31]. However, it is inconvenient to obtain experimental camels, and, generally, the FaeG proteins expressed in prokaryotic systems typically form inclusion bodies, resulting in the loss of their natural conformation and functional activity [32,33]. Herein, we achieved the soluble expression of the FaeG protein in a prokaryotic system. Additionally, we presented a rapid and efficient single-B-cell sorting method designed for screening neutralizing antibodies against porcine ETEC from the memory B-cell pool of mice immunized with the target antigen.

And the sorting of memory B cells specific for rFaeG represents a pivotal step. Consequently, the quality of the antigen protein is of paramount importance, directly influencing the efficacy of the selected variable regions and subsequent optimizations. However, owing to the inclusion body problem of prokaryotic expression, antigen protein expression necessitates specialized in vitro renaturation conditions, with renatured proteins potentially encountering issues such as misfolding and insolubility [32]. Conversely, soluble expression yields relatively pure protein samples, thereby simplifying subsequent purification steps. To date, the expression of the FaeG protein has been studied using various systems, including fimbrial shearing, mammalian cells, and plant expression systems [12,34,35]. Furthermore, Lu et al. utilized multi-epitope fusion antigen (MEFA) technology to segment the FaeG protein sequence into nine segments, which were fused with the CfaB protein to achieve soluble expression in a prokaryotic system. They identified six segments as neutralizing epitopes [36]. The presence of non-target protein segments expressed through fusion may result in false positives during mouse immunization, thereby complicating the screening process for variable regions of neutralizing antibodies against the FaeG protein. To address this, our study employed molecular chaperones to facilitate proper protein folding [37]. By utilizing the pGro7 plasmid, which expresses the molecular chaperones GroEL and GroES, we successfully achieved the soluble expression of the rFaeG protein. Subsequently, the rFaeG protein was purified via nickel ion affinity chromatography. This accomplishment lays the groundwork for detecting anti-FaeG antibodies in serum and for isolating anti-FaeG-specific memory B cells using fluorescence-activated cell sorting technology.

Because of the challenges in expressing the FaeG protein in prokaryotic systems, no subunit vaccine targeting the full-length FaeG protein is currently available. Moreover, it is not possible to determine whether the soluble rFaeG protein has the correct native structure in a short period of time. Therefore, it is necessary to select a suitable immunogen. However, formaldehyde is not only a potent cross-linking agent that effectively immobilizes bacterial proteins and inhibits bacterial growth and replication, as, during the inactivation process, the surface antigen structure of the bacteria remains partially intact, preserving some degree of the natural structure of the bacterial outer wall and, thus, maintaining immunogenicity [38,39]. This approach was used with the aim of inducing the production of neutralizing antibodies against the FaeG protein present on the surface of ETEC. Moreover, iELISA was employed to assess serum IgG titers in immunized mice, using ETEC and rFaeG as detection agents. The findings revealed a synchronous and continuous increase in mouse serum IgG titers detected using both ELISA methods, validating the successful expression of the soluble rFaeG protein in this experiment. Consequently, employing rFaeG as a screening antigen for cell sorting ensures the reliable identification of the variable regions of neutralizing antibodies.

In the spleen of mice, non-B-lymphocyte immune cells expressing FcγR are present and can capture some IgG memory B cells, potentially influencing the final screening outcomes [40]. Starkie et al. addressed this issue by using FcγR-blocking reagents to obstruct FcγR on the cell surface [41]. Subsequently, they targeted CD45R to enrich B cells and used fluorescently labeled antibodies against CD19, IgG, IgM, and IgD to sort IgG memory B cells, achieving a final positive sorting rate of 38.5% (ratio of antigen-specific recombinant antibody production by sorted single cells) [41]. Conversely, Prashar et al. exclusively used IgG as the labeling target. While their method, involving seamless fusion cloning post-cell sorting, proved efficient, rapid, and cost-effective, it exhibited a lower positive detection rate of 14.5% [42]. Although employing a broader range of antibodies for sorting enhances the accuracy of targeting B cells, it also increases the associated expenses. Conversely, using fewer antibody types increases the likelihood of sorting false-positive B cells. Therefore, our study employed CD19 and IgG fluorescently labeled antibodies for the dual sorting of IgG memory B cells, followed by the use of rFaeG labeled with two bright fluorescent groups, APC and PE, to screen for antigen-specific IgG memory B cells. Additionally, we utilized degenerate primers for murine IgG light- and heavy-chain antibody variable regions, as described by Kettleborough in 1993, for seamless fusion and cloning into pcDNA3.4, which contains the murine IgG Fc fragment, facilitating the expression of recombinant antibodies [24]. This approach yielded a final positive sorting rate of 36.2% (21/58), without evident drawbacks in sorting efficiency. By minimizing the types of antibodies used, we not only reduced costs, but also simplified and facilitated the screening process [43,44].

In this study, we used mice as experimental models to conduct preliminary experimental studies on anti-rFaeG mAbs. Once the efficacy and safety of these antibodies are confirmed in the mouse model, researchers can use antibody-derivative technology in pig models. This approach will help adapt antibody therapeutics to the complex in vivo and ex vivo environments of clinical settings and aid in better understanding their potential applications in real-world scenarios.

An intraperitoneal injection of ETEC allows ETEC to traverse the intestinal membrane into the lumen [45]. The ileum is responsible for the immune function of the small intestine. Thus, changes in ileal tissue can reveal the extent of intestinal inflammation. Severe intestinal inflammatory responses were observed in the group 1 mice following the intraperitoneal ETEC injection, as demonstrated by IHC and HE-stained tissue sections, indicating a robust activation of the immune response within the peritoneal cavity. The IHC sections showed significantly higher ETEC-positive brown–yellow staining in group 1 than in groups 2 and 3. This suggests that anti-rFaeG mAbs effectively neutralized ETEC within the peritoneal cavity, reducing the pathogen load and thereby mitigating the inflammatory response and tissue damage in the intestines, ultimately improving the survival rate of the mice.

## 5. Conclusions

In conclusion, we refined a method for screening the variable region of target antibodies from a mouse memory B cell library and applied it to the preliminarily production of porcine ETEC antibody therapeutics. This study successfully generated a recombinant mouse IgG antibody with neutralizing activity. Therefore, further research on antibody derivatives of our monoclonal antibody against the pathogen can be carried out, with a view to application in clinical environments. Additionally, the successful establishment of an antibody library and the selection of highly specific antibodies further support the potential application of antibody-based therapeutics as alternatives to antibiotics in veterinary medicine. This progress lays the groundwork for developing antibody drugs targeting the F4ac subtype of the FaeG protein to combat enterotoxigenic *E. coli* diarrhea in piglets, thereby advancing efforts to mitigate economic losses and reduce the overuse of antibiotics in agriculture.

## Figures and Tables

**Figure 1 vetsci-11-00419-f001:**
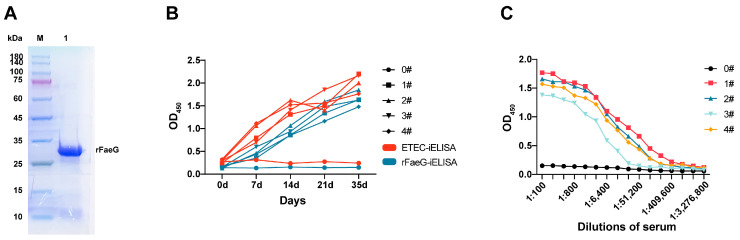
SDS-PAGE analysis of rFaeG soluble expression and the detection of immunized mouse serum. (**A**) SDS-PAGE analysis of expressed and purified rFaeG in the supernatant. Lane M: protein ladder marker; Lane 1: purified rFaeG. (see Appendix A) (**B**) Growth trend of serum antibodies detected using two iELISA methods. (**C**) Serum IgG titers of four immunized mice. Serum of 0# used as negative control.

**Figure 2 vetsci-11-00419-f002:**
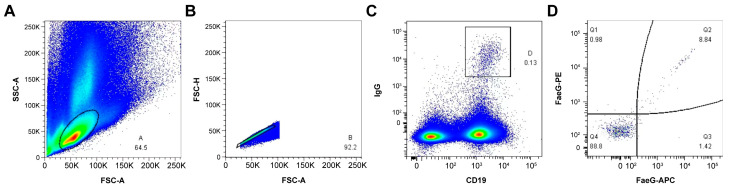
The sorting of rFaeG-specific mouse memory B cells using FACS. (**A**) B-lymphocyte population was gated and identified using forward scatter area (FSC-A) and side scatter area (SSC-A) parameters (events account for 64.5%). (**B**) FSC-A and FSC-height (FSC-H) density plot was used to exclude doublets and separate from debris (events account for 92.2%). (**C**) The memory B-cell subsets were isolated by CD19 and IgG gates (events account for 0.13%). (**D**) The rFaeG-specific memory B cells were gated by APC and PE fluorescently labeled rFaeG (events account for 8.84%). Crimson: AF700 dye fluorescence excitation; Green: FITC dye fluorescence excitation; Red: APC dye fluorescence excitation; Orange red: PE dye fluorescence excitation; Blue: No labeled cells were activated by fluorescence.

**Figure 3 vetsci-11-00419-f003:**
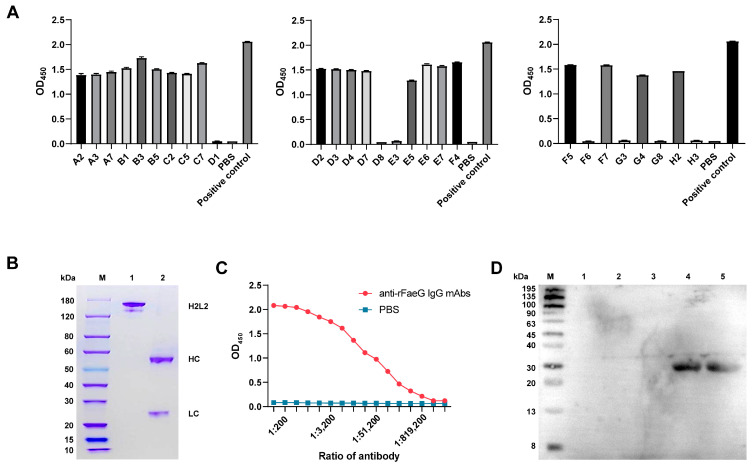
The expression of anti-rFaeG IgG mAbs. (**A**) The binding ability of 28 PCR-positive wells corresponding to the expression of antibodies in response to rFaeG was detected using iELISA. (**B**) SDS-PAGE analysis. Lane M: protein ladder marker; Lane 1: the H2L2 antibody of the mAbs; Lane 2: the light and heavy single chains (see Appendix A). (**C**) The affinity detection of anti-rFaeG IgG mAbs. We diluted the mAbs from 1:100 to 1:3,276,800 and used PBS as a control. (**D**) The specificity detection of anti-rFaeG IgG mAbs using Western blotting. Lane M: protein ladder marker; Lane 1: DH5α; Lane 2: BL21; Lane 3: non-F4-type *E. coli*; Lane 4: bacterial cells of ETEC; Lane 5: rFaeG.

**Figure 4 vetsci-11-00419-f004:**
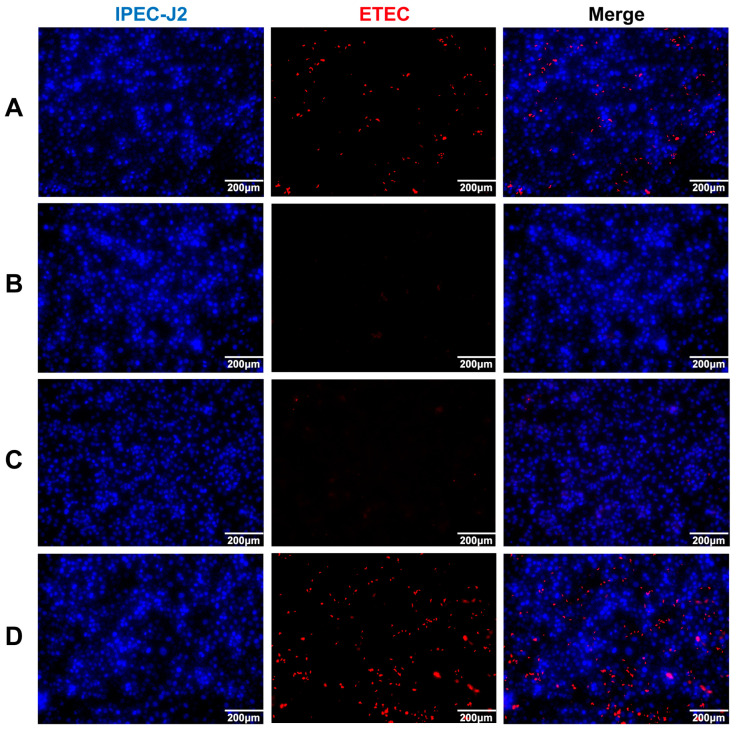
The suppression effect of anti-rFaeG IgG mAbs on 3 × 10^8^ CFU/mL ETEC. (**A**) ETEC incubated with negative serum. (**B**) ETEC incubated with positive serum, obtained from mouse 1#. (**C**) ETEC incubated with anti-rFaeG IgG mAbs. (**D**) ETEC incubated with PBS. Red: Dylight 594 dye fluorescence activation indicates ETEC; DAPI dye fluorescence activation indicates IPEC-J2.

**Figure 5 vetsci-11-00419-f005:**
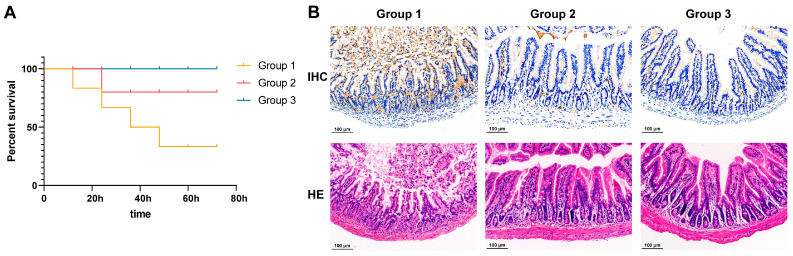
The anti-rFaeG mAb neutralization assay in mice. (**A**) Survival curves of the three groups. (**B**) Ileal tissue sections were stained with IHC and HE. Group 1: injection of ETEC directly into the peritoneal cavity. Group 2: injection of ETEC incubated with mAbs. Group 3: injection of sterile PBS into the peritoneal cavity as a control.

**Table 1 vetsci-11-00419-t001:** Multiplex and nested PCR primer sequences for V_H_ and Vκ.

Primers	Sequence (5′-3′) *^a^*
V_H_1-F	(CCGATCCAGCCTCCGGAC)MAGCTTCAGGAGTCRGGACC
V_H_2-F	(CCGATCCAGCCTCCGGAC)CAGCTGAAGSASTCAGGACC
V_H_3-F	(CCGATCCAGCCTCCGGAC)MWGSKGGTGGAGTCTGGGGGA
V_H_4-F	(CCGATCCAGCCTCCGGAC)ARSSTGGWGGAATCTGGAGGA
V_H_5-F	(CCGATCCAGCCTCCGGAC)ARGSTGRTSGAGTCTGGAGG
V_H_6-F	(CCGATCCAGCCTCCGGAC)CARSYGCAGCARYCTGGG
V_H_7-F	(CCGATCCAGCCTCCGGAC)CAGYTGSWGCARTCTGGA
V_H_8-F	(CCGATCCAGCCTCCGGAC)CAGCTGCAGCAGTCWGTG
V_H_9-F	(CCGATCCAGCCTCCGGAC)MASYTGSWGGWGWCTGGAGG
V_H_10-F	(CCGATCCAGCCTCCGGAC)CAGMTSCAGCAGYCTGG
1st reverse γ	CKYGGTSYTGCTGGCYGGGTG
2nd reverse γ	GCTCAGGGAARTAGCCCTTGAC
Vκ1-F	(CCGATCCAGCCTCCGGAC)CAGACRTCMAGATRAYCCAGWCTMCA
Vκ2-F	(CCGATCCAGCCTCCGGAC)CARAMATTKTGCTGACYCARTYTCC
Vκ3-F	(CCGATCCAGCCTCCGGAC)CAGATRYTKTGATGACCCAAACTCCA
Vκ4-F	(CCGATCCAGCCTCCGGAC)CASRAAWTSTTCTCWYMCAGTGTCC
1st reverse κ	CTAACACTCATTCCTGTTGAAGC
2nd reverse κ	TGGGAAGATGGATACAGTT

V_H_: variable region of heavy chain; Vκ: variable region of kappa light chain. *^a^* The upstream homology arms (the sequences in parentheses) were added on the basis of the reference primers, and the sequences of the reverse primers were the downstream homology arms.

## Data Availability

The original contributions presented in the study are included in the article/Appendix A; further inquiries can be directed to the corresponding author.

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
