# Peer review of "Screening of Neutralizing Antibodies against FaeG Protein of Enterotoxigenic Escherichia coli"

_vetsci, 2024, doi:10.3390/vetsci11090419_

Round 1

Reviewer 1 Report

Comments and Suggestions for Authors

In this paper, the authors immunized ETEC and successfully established anti-FaeG antibodies by screening with recombinant FaeG. The FaeG antibodies showed neutralizing activity against ETEC in vivo and in vitro. The methodology is appropriate and there are no major problems with the results. However, there are some inadequacies in the text, which should be corrected.

My comments are as follows;

1.    In Fig.1A, the authors present the results of expression checks for soluble and insoluble, with and without arabinose induction. However, reading the text, I do not understand what they are giving this data for. Figure 1A should be removed and only Figure 1B presented showing that recombinant FaeG could be prepared from the soluble fraction.

2.    In section 2.2, preparation of recombinant FaeG, there is a lack of description of how the protein was prepared from the recovered E. coli. Was no bacterial disruption performed?

3.   Figures 3-6 in the text (for example, Figure 3 on line 253) are probably a misnomer for Figures 2-5. It should be checked and corrected.

4.    I couldn't understand the explanation for the figure legend in Figure 3A. Isn't this the ELISA data for the 28 antibodies obtained in the screening?

5.    Figure 3B suddenly has something called H2L2, but there is no explanation anywhere; it should be explained from which well data was chosen for the results in Figure 3A.

6.    In Figure 3D, the specificity of the antibody for recombinant FaeG is evaluated, but the possibility that it recognizes the His-tag cannot be ruled out.

7.   The explanations in Figures 4 and 5 are not clear, so please write the explanations in the figures.  For example, Figure 4A says negative serum, Group 1 in Figure 5 says injected ETEC only…

Reviewer 2 Report

Comments and Suggestions for Authors

This paper discusses “Screening of neutralizing antibodies against FaeG protein of enterotoxigenic Escherichia coli”. In general, the results of the paper confirm the rationality and correctness of the experimental method, which is consistent with the research goal. However, there are still some questions that hope the author to answer. Some comments or suggestions are as follows:

1. What is the method for establishing the experimental model in this study?

2. The paper mentioned that enterotoxin Enterotoxigenic Escherichia coli (ETEC) is the main pathogen of pig E. coli related diarrhea, it sounds very interesting, but the experimental animals in this study are mice, can you say why not pigs?

3. How to select the F4ac FaeG protein in ETEC? 

4. How to determine whether ETEC causes damage to the mouse ileum?

5. The picture described in result 3.3/3.4 does not seem to correspond, please check carefully.

Reviewer 3 Report

Comments and Suggestions for Authors

Yang Tian et al. submitted a MS titled “Screening of neutralizing antibodies against FaeG protein of enterotoxigenic Escherichia coli1”. The contents seems to have a high potential in terms of using an antibodies instead of antibiotics. The authors tested the effect with mice that might be easy to handle but not the animal occurring the symptoms. The objective animal might be different environment in terms of pH, microbioum etc. They used antibodies from mice which were so much expensive in reality. So the authors should give explanation in terms of antibodies and objective animals.   

Comments on the Quality of English Language

need to be read native speaker

Reviewer 4 Report

Comments and Suggestions for Authors

The submitted manuscript:  Tian et al., 'Screening of neutralizing antibodies against FaeG protein of 2 enterotoxigenic Escherichia coli'- is an interesting paper  that explores the development of mAb as alternative to antibiotic therapy  for  the treatment of ETEC in swine.

 Although many important concepts (ie. cell sorting with selected antibody markers, IP injection of ETEC  as a challenge model and so forth) were adequately addressed with the discussion; in this reviewer’s opinion, some other items should be further adressed  prior to publication.

Line 91: How was the amount of administered ETEC ( 4 x108 CFU/ml) determined for immunization- was this concentration optimal for antibody production. Either add a supporting refence, or discuss the  possibility of a sub-optimal amount of ETEC and reduce antibody production.

Line 299 (Figure 4): The figure legend is unclear. The authors should add further details describing the figures panels.

Line 311-315, 410. In the manuscript version this reviewer received- there is no Figure 6 within either  the paper or  supplemental figures. IHC appears with Figure 5 in the supplemental files.

Comments on the Quality of English Language

The manuscript is well written. There are some grammar and sentence structure edits that are needed. These are minor however, and should be identified (and corrected) during the final revision of the paper

Round 2

Reviewer 4 Report

Comments and Suggestions for Authors

The comments associated  with the initial manuscript review were addressed